# A Systematic Review of Inter-Brain Synchrony and Psychological Conditions: Stress, Anxiety, Depression, Autism and Other Disorders

**DOI:** 10.3390/brainsci15101113

**Published:** 2025-10-16

**Authors:** Atiqah Azhari, Ashvina Rai, Y. H. Victoria Chua

**Affiliations:** Psychology Programme, School of Humanities and Behavioural Sciences, Singapore University of Social Sciences, Singapore 599494, Singapore; ashvinarai@suss.edu.sg (A.R.);

**Keywords:** inter-brain synchrony (IBS), interpersonal neural synchronization, brain-to-brain coupling, hyperscanning, social neuroscience, psychological disorders, autism spectrum disorder (ASD), anxiety, depression

## Abstract

Background: Inter-brain synchrony (IBS)—the temporal alignment of neural activity between individuals during social interactions—has emerged as a key construct in social neuroscience, reflecting shared attention, emotional attunement, and coordinated behavior. Enabled by hyperscanning techniques, IBS has been observed across a range of dyadic contexts, including cooperation, empathy, and communication. This systematic review synthesizes recent empirical findings on inter-brain synchrony (IBS)—the temporal alignment of neural activity between individuals—across psychological and neurodevelopmental conditions, including stress, anxiety, depression, and autism spectrum disorder (ASD). Methods: Drawing on 30 studies employing hyperscanning methodologies (EEG, fNIRS, fMRI), we examined how IBS patterns vary by clinical condition, dyad type, and brain region. Results: Findings indicate that IBS is generally reduced in anxiety, depression, and ASD, particularly in key social brain regions such as the dorsolateral and medial prefrontal cortices (dlPFC, mPFC, vmPFC), temporoparietal junction (TPJ), and inferior frontal gyrus (IFG), suggesting impaired emotional resonance and social cognition. In contrast, stress elicited both increases and decreases in IBS, modulated by context, emotional proximity, and cooperative strategies. Parent–child, therapist–client, and romantic dyads exhibited distinct synchrony profiles, with gender and relational dynamics further shaping neural coupling. Conclusions: Collectively, the findings support IBS as a potentially dynamic, condition-sensitive, and contextually modulated neurophysiological indicator of interpersonal functioning, with implications for diagnostics, intervention design, and the advancement of social neuroscience in clinical settings.

## 1. Introduction

Inter-brain synchrony (IBS), also known as interpersonal neural synchronization, refers to the alignment or temporal coupling of neural activities between two individuals engaged in social interactions or shared tasks [1,2]. Utilizing hyperscanning methodologies with electroencephalography (EEG), functional near-infrared spectroscopy (fNIRS), and functional magnetic resonance imaging (fMRI), recent research has provided substantial evidence that individuals exhibit synchronized neural responses in real time during cooperative interactions, empathetic exchanges, or emotional attunement [3,4,5]. Such synchrony has been observed prominently in brain regions associated with social cognition, emotional processing, and executive control, including the prefrontal cortex, temporoparietal junction, and superior temporal sulcus [6,7].

Importantly, IBS is not a binary phenomenon. Measures of IBS quantify the magnitude of neural coupling between individuals, and the degree of IBS may fluctuate depending on factors such as task demands, the nature of the dyadic relationship, individual traits, and moment-to-moment engagement [8,9]. These fluctuations reflect the complex and adaptive nature of social interaction. Given that IBS reflects a continuous and context-sensitive neural state, it offers a promising window into the neural dynamics of social interaction.

A growing body of evidence suggests that IBS is a core neural mechanism supporting social communication and cooperation [10,11,12]. For instance, Kinreich [13] found enhanced IBS in the temporoparietal junction (TPJ) among couples compared to strangers, pointing to the influence of relationship familiarity on neural coupling. Another study by Cañigueral [14] reported increased IBS during effective information sharing between conversation partners, particularly in the TPJ. In an fMRI study, speaker-listener neural coupling in language-related regions facilitated mutual understanding, and this synchrony diminished when communication was disrupted [15]. Likewise, a recent review revealed patterns of increased IBS in the prefrontal cortex (PFC) during collaborative tasks compared to when individuals worked alone [16]. Collectively, these findings underscore the associations between IBS and shared attention, nonverbal coordination and successful interpersonal communication.

However, it remains an open question as to whether IBS is a causal mechanism facilitating social connection, or merely a neural byproduct of successful interaction. Disentangling these interpretations is especially important when considering IBS as a potential biomarker in clinical contexts. Psychological conditions such as stress, anxiety, and depression, are known to impair socio-emotional functioning and may disrupt the neural processes that facilitate effective interpersonal communication and social attunement. In addition, some neurodevelopmental disorders, such as autism spectrum disorder (ASD), are characterized by atypical socioemotional development and communication patterns, which may reflect or contribute to marked changes in IBS. As such, there is growing interest in whether IBS can be extended beyond normative interpersonal processes to serve as a marker of disrupted interpersonal functioning in psychological (stress, anxiety and depression) and neurodevelopmental conditions (ASD).

Two foundational neurobiological models, the Social Brain Network (SBN) and the Mirror Neuron System (MNS), provide a theoretical basis to contextualize IBS as a potential diagnostic and therapeutic marker. The SBN comprises regions such as the medial prefrontal cortex (mPFC), TPJ, and superior temporal sulcus (STS), which are implicated in mentalizing and theory of mind processes [17]. IBS observed via hyperscanning often involves these regions during tasks requiring joint attention or affective resonance. Variations in synchrony within the SBN may correspond to differences in social cognitive functioning across individuals, including those with autism spectrum traits or clinical diagnoses. Similarly, the MNS, particularly the inferior frontal gyrus (IFG) and inferior parietal lobule (IPL), is posited to support embodied simulation, enabling individuals to internally model others’ actions and intentions [18]. Fluctuations in mu-rhythm synchrony across these regions during cooperative or imitative tasks may reflect individual differences in motor resonance and affective alignment [19]. These patterns, while not necessarily pathological, may help characterize atypical interpersonal processing found in certain psychological or neurodevelopmental conditions. Thus, IBS patterns anchored in these systems offer clinically relevant insights while remaining sensitive to contextual modulations in social interaction.

Despite significant advancements in the study of inter-brain synchrony (IBS), several critical gaps remain in understanding its relationship with psychological and neurodevelopmental conditions. First, while IBS has been observed across a range of interpersonal contexts, no systematic synthesis has yet compared how different dyad types—such as parent–child, therapist–client, romantic partners, or unfamiliar adult pairs—uniquely modulate patterns of neural synchrony in clinical populations. This is particularly important because the nature of the relationship may influence the degree and quality of synchrony, with emotionally bonded dyads (e.g., parents and children) potentially exhibiting different IBS dynamics than professional or unfamiliar interactions (e.g., therapist-client or peer dyads).

Second, there is a lack of integrated understanding of how specific psychological conditions interact with different dyadic contexts to shape IBS. For example, maternal anxiety may disrupt parent–child synchrony differently compared to how peer-based anxiety affects adolescent interactions. Similarly, children with ASD may show distinct IBS patterns when engaging with parents compared to peers or clinicians. These nuanced interactions between condition type and dyadic relationship have yet to be systematically explored.

Third, while certain brain regions—such as the dorsolateral PFC (dlPFC), medial and ventromedial prefrontal cortices (mPFC/vmPFC), TPJ, and IFG—are frequently implicated in IBS research, it remains unclear whether these regions function as universal hubs for social neural coupling or if different psychological disorders exhibit condition-specific patterns of regional synchrony. A more precise mapping of consistent versus disorder-specific brain region involvement is needed to clarify whether particular neural substrates are broadly related to interpersonal functioning or uniquely tied to specific emotional or cognitive impairments.

To address these gaps, the present systematic review synthesizes empirical evidence from the past decade, explicitly examining IBS between dyadic pairs in relation to stress, anxiety, depression, autism, and related disorders. The review addresses two primary research questions: (1) Can alterations in IBS reflect underlying psychological conditions, thereby potentially serving as a neurophysiological biomarker? (2) Which specific brain regions are consistently involved in IBS across these psychological and neurodevelopmental conditions? Addressing these questions will offer important insights into interpersonal neural dynamics, refine theoretical models of social neuroscience, and inform clinical approaches, including diagnostics and interventions aimed at enhancing social functioning in affected populations.

## 2. Methods

This systematic review was conducted in accordance with PRISMA guidelines to synthesize empirical research examining inter-brain synchrony (IBS) in the context of psychological and neurodevelopmental conditions. Two electronic databases, PubMed and PsycInfo, were systematically searched at two time points (July 2024 and July 2025) to identify peer-reviewed articles published in the past decade, with a timeframe from 2014 to 2024.

The search strategy employed a combination of terms related to IBS and specific psychological conditions. The core search string included multiple variations of the term “inter-brain synchrony” (e.g., Interbrain Synchrony, Brain-to-brain Synchrony, Intersubject Synchronization, Interpersonal Neural Synchronization, Hyperscanning) in combination with condition-specific keywords: stress, anxiety, depression, autism, and social disorder Five distinct queries were run for each condition using Boolean operators (refer to Appendix A). The literature search and initial screening were conducted by the first and second author. Duplicate records were identified and removed manually prior to full-text screening.

Studies were only eligible for inclusion if they met the following criteria: (1) published in English peer reviewed journals between 2014 and 2024; (2) involved human participants; (3) employed hyperscanning methods (e.g., EEG, fNIRS, fMRI) to measure IBS in dyads, and (4) assessed IBS in the context of a psychological or neurodevelopmental condition. In addition to database searches, reference lists of eligible articles were also screened to identify additional relevant studies. Studies were excluded if they (1) did not involve hyperscanning or dyadic IBS measurements; (2) focused on animal models or pharmacological interventions; or (3) did not examine any psychological or neurodevelopmental condition.

Data extracted from the included studies encompassed: type of dyad (e.g., parent–child, patient–clinician), neuroimaging method used, sample size (number of dyads), the specific psychological condition examined, and the brain regions implicated in IBS. Figure 1 illustrates the study selection process using a PRISMA flowchart.

## 3. Results

A total of 30 studies were included in the review. The studies are categorized across five primary psychological and neurodevelopmental conditions: autism spectrum disorder (ASD) (N = 11), stress (N = 10), anxiety (N = 5), depression (N = 2), and other disorders (N = 5) (see Table 1, Table 2, Table 3, Table 4 and Table 5). Four studies investigated more than one condition; therefore, the total number of studies categorized by condition does not sum to the overall number of studies reviewed.

Figure 2 illustrates the types of neuroimaging hyperscanning techniques employed—primarily fNIRS, EEG, or fMRI—to measure IBS across various dyadic configurations. Overall, across 30 included studies, our findings reveal that fNIRS is the most commonly used neuroimaging tool. Figure 3 outlines the types of dyads in each study, categorized by condition. Parent–child interactions are the most frequently studied, reflecting a research focus on family dynamics in psychological conditions.

Table 1 summarizes a breakdown of all studies by conditions. Among all the conditions, ASD was the most studied condition (N = 11 studies). Studies predominantly used fNIRS (N = 9 studies), with the remaining using EEG (N = 2 studies). Of the 11 studies, five studies included parent–child dyads, indicating a particular interest in understanding ASD and IBS within family interactions. Three studies on ASD investigated differences in IBS between dyads with varying levels of autistic traits (such as high–high, low–low, and high–low pairings) indicating a research interest in exploring how similarity or mismatch in autistic traits influences IBS [20,21,22]. Notably, the comparison of matching versus mismatching pairs is unique to studies in autism included in this review. This approach likely draws on the social interaction mismatch hypothesis [23] and double empathy problem [24,25], which proposes that interpersonal dissimilarity can disrupt synchronization processes. The remaining autism studies examined IBS during interactions between individuals with autism and healthy controls or confederates (N = 3).

Stress was the second most studied condition (N = 10 studies). All studies used fNIRS as the neuroimaging technique. The most common dyad type was the parent–child dyad (N = 5 studies), reflecting a particular interest in how caregiving-related stress augment interbrain synchrony and subsequently affects caregiving relationships. Due to the relative ease with which stress can be experimentally induced in controlled settings (compared to other conditions included in this review, such as depression or anxiety), several studies examining stress included cooperative pair adult-adult dyads (N = 3 studies), in which two healthy adults were paired to complete collaborative tasks. This design allowed researchers to examine IBS during acute stress responses in healthy individuals, providing a valuable model for understanding stress-related neural dynamics without the need for clinical population. The remaining two studies included spousal dyads and patient–patient dyads (N = 1 each).

A total of five studies focused on anxiety. Two studies used EEG, two studies used fNIRS, and one study used fMRI. Of the five studies, two of them involved cooperative pair adult–adult dyads, two of them involved parent–child dyads, and one study involved patient–clinician dyads. Depression was the least studied among our selected conditions (N = 2 studies), with one study each using fMRI and fNIRS. Notably, these studies were a subset of the studies focused on anxiety and involved parent–child dyads (N = 1 study) and cooperative pair adult–adult dyads (N = 1 study).

Finally, five studies investigated psychopathological conditions outside of stress, anxiety, depression, and autism spectrum disorder. These “other” conditions included clinical high risk for psychosis, alcohol use disorder, gaming disorder, borderline personality disorder, and aggressive or rule-breaking behaviors. These studies used fMRI and fNIRS, with a focus on dyads comprising patients and healthy controls, as well as parents and children.

**Table 1 brainsci-15-01113-t001:** Summary Table of Records Across All Conditions.

Condition	No. of Studies	Dyad Types	Devices Used	Common Brain Regions Involved	Total Sample Size (Pairs)
Autism Spectrum Disorder	11	Patient–Clinician, Patient–Confederate, Parent–Child, Child–Child, Adult–Adult	lfNIRS, lEEG	mPFC, TPJ, LPFC, SFC, Fronto-temporo-parietal network	465
Stress	10	Patient-Patient, Parent–Child, Spousal partners, Adult–Adult	lfNIRS	PFC (dlPFC, FPC, OFC), SFG, TPJ, Wernicke’s, rTPJ	561
Anxiety	5	Patient–Patient, Patient–Clinician, Patient–Confederate, Parent–Child, Adult–Adult	lEEG, lfNIRS, lfMRI	dlPFC, TPJ, mPFC, Amygdala, Broca’s, Wernicke’s	480
Depression	2	Parent–Child, Adult–Adult	lfNIRS, lfMRI	dlPFC, vmPFC, OFC, Amygdala, Hippocampus, TPJ	415
Other Disorders (e.g., BPD, CHR)	5	Patient–Patient, Patient–Healthy Controls, Parent–Child	lfMRI, lfNIRS	TPJ, IFG, mPFC, vlPFC, dlPFC, Hippocampus	569

### 3.1. Inter-Brain Synchrony and Autism Spectrum Disorders (ASD)

Across the reviewed studies, IBS in individuals with ASD revealed both impairments and preserved capacities, depending on the social context, task demands, and age group. Table 2 summarizes the findings relating to IBS in ASD conditions. Across 11 studies, IBS in ASD was most consistently examined in prefrontal and temporoparietal regions, showing reduced synchrony during socially demanding tasks (e.g., conversations, cooperation) compared to typically developing (TD) controls [26,27,28,29,30].

Overall, IBS is generally found to be reduced in ASD populations but with anomalies pointing to potential compensatory mechanisms under specific conditions. Key anomalies included enhanced IBS in the superior temporal gyrus (STG) and frontopolar regions within high-trait adult dyads despite atypical behaviors [20], and greater IBS in the right inferior parietal lobule (r-IPL) in autistic compared to typical children during specific imitation tasks [31]. In fact, IBS was observed to be similar or enhanced compared to TD controls in supportive contexts across several studies. Specifically, increased IBS was reported when parents of children with ASD adapted their response patterns to better match their children [28], during music therapy sessions (compared to storytelling) [32] and meaningful gesture imitation (compared to meaningless gesture imitation) [31].

#### 3.1.1. Inter-Brain Synchrony in Typical Development vs. ASD

Research contrasting IBS in TD individuals with those diagnosed with ASD has provided insights into how ASD impacts social neural processes. TD individuals generally exhibit robust IBS during social interactions, facilitating efficient social communication, cooperation, and understanding [10,11,12]. A number of studies observed that individuals with ASD commonly show atypical IBS patterns during social interactions.

For example, Quiñones-Camacho [27] examined IBS during natural conversational interactions between adults with ASD and the experimenter. Compared to TD controls, adults with ASD showed reduced synchronization in the TPJ. The reduction in IBS correlated significantly with deficits in social communication abilities. As a crucial hub for social cognition and mental state inference, the TPJ appears compromised in individuals with ASD, impacting their capacity to dynamically coordinate with others during interactions. Similarly, Key [33] found that lower levels of IBS during conversations between an experimenter and adolescents with autism were associated with greater behavioral symptoms of social difficulties, further supporting the link between neural synchrony and social functioning in ASD. This relationship was notably more robust in female participants, indicating potential sex-specific neurological differences in social synchronization mechanisms. Such findings highlight the importance of considering gender when evaluating social competence and designing targeted therapeutic interventions for ASD.

In parallel, Du [31] demonstrated that IBS outcomes depend on task relevance. In adult–child dyads, children with autism exhibited higher IBS in the right IPL (r-IPL) during imitation of meaningful gestures, but lower IBS during imitation of meaningless gestures, compared to their TD peers. IBS in the r-IPL during meaningless gesture imitation significantly predicted the likelihood of autism. These findings underscore the potential of IBS to reflect impairments in real-time social coordination in ASD.

In contrast, investigations into cooperative tasks involving parent–child dyads reveal nuances in IBS patterns modulated by task meaning and dyad type. Wang [30] found increased frontal cortex synchronization in children with ASD when engaging cooperatively with their parents compared to working alone. However, this synchronization was inversely correlated with the severity of autism symptoms, indicating that greater symptom severity was associated with weaker IBS during cooperation. Kruppa [26] studied a younger cohort of children with ASD and observed reduced motor synchrony but no significant differences in IBS using similar cooperative tasks in [30]. Taken together, these findings suggest that task context, age, and developmental factors may also influence the nature of IBS alterations in ASD.

#### 3.1.2. Parental Influence on Inter-Brain Synchrony in ASD

Importantly, recent research has also begun to emphasize the significant role of parental behavior in shaping IBS patterns within ASD. Specifically, burgeoning evidence suggests that parental behaviors play a compensatory role, helping to enhance IBS during interactions with their children. A study [28] examined IBS in parent–child dyads during cooperative tasks and found that children with ASD displayed comparable synchronization levels to TD peers. This was largely due to adaptive behaviors exhibited by their parents, who adjusted their behaviors to closely align with their children’s rhythms and patterns, effectively compensating for the children’s synchronization difficulties. Further extending this work, another study [32] introduced a neurologic music therapy protocol with ASD child–parent dyads, observing enhanced IBS in the right IFG and frontopolar regions during emotionally engaging musical interactions than storytelling. Notably, synchronization was predominantly directed from parent to child, suggesting that the neural activity of the parent tended to influence that of the child. Similarly, Minagawa [29] explored IBS in very young infants and mothers during breastfeeding, a fundamental attachment behavior. They found no significant group differences, observing strong IBS between the mother’s ventral PFC, especially in the right orbitofrontal cortex (OFC) and the infant’s bilateral TPJs, across both TD infants and infants at risk for ASD. Interestingly, infants at risk for ASD showed stronger IBS with their mothers during breastfeeding compared to holding, specifically between the mother’s left middle temporal gyrus (L-MTG) and the infant’s left STG (L-STG). This effect was not observed in TD infants. The increased synchronization was positively correlated with maternal emotional sensitivity, suggesting that mothers of ASD-risk infants, who may be more attuned due to prior parenting experiences, played a key role in driving the neural coupling. This specific synchronization highlights crucial neural substrates linked to reward processing and primitive mentalization processes in infants.

These findings suggest that while ASD is associated with disrupted neural synchrony at the individual level, supportive social contexts, like adaptive parenting or engaging therapeutic activities, can modulate IBS and potentially improve social attunement. Thus, studies on parent–child dyads extend the understanding of IBS in ASD by demonstrating how interpersonal dynamics and task relevance shape neural synchronization, offering avenues for intervention.

#### 3.1.3. Matching Autistic Traits and Inter-Brain Synchrony

Beyond group-level comparisons, some studies have investigated whether interpersonal similarity in autism-related traits influences IBS. These studies typically formed dyads based on trait-level matching: pairing individuals with high–high, low–low or mismatched high–low levels of autistic traits, to examine whether matching symptom profiles may facilitate greater neural synchrony during interaction.

Using fNIRS hyperscanning, Peng [20] found that dyads consisting of college students with high autistic traits exhibited atypical behavioral patterns during interactions, including reduced mutual gaze and fewer conversational exchanges. However, these dyads demonstrated higher IBS levels, suggesting effective, though unconventional, communication styles. This finding aligns with the “double empathy problem” [24,25] which posits that communication issues between autistic and neurotypical individuals result from reciprocal difficulties rather than unilateral social deficits within the autistic individual.

However, not all findings align with this view. Feng [22] reported that only dyads with low–low (LL) autistic traits exhibited significant IBS in the right TPJ (rTPJ) during conversation. In contrast, dyads with high–high (HH) autistic traits showed significantly lower rTPJ IBS than LL dyads, while high–low (HL) dyads did not differ from either group. Moreover, across all dyads, higher average autistic traits were linked to reduced rTPJ IBS. These findings suggest that rTPJ synchrony during conversation is attenuated with increasing autistic traits regardless of dyadic matching, consistent with rTPJ’s central role in mentalizing and perspective-taking.

Extending this line of inquiry, Zhou [21] examined IBS modulation through autistic traits in school-aged children’s peer interactions. Across three structured tasks that varied in social complexity (watching videos, shared reading, playing Jenga), they found that the relationship between autistic traits and IBS in the frontotemporoparietal networks depended both on task demands and trait profiles within dyads. During video watching, greater increases in IBS were associated with higher autistic traits and larger differences in traits between the children. In contrast, during shared reading, IBS in the frontotemporoparietal networks was reduced when both children had elevated or similar levels of autistic traits. During the highly interactive Jenga task, no significant associations were found between IBS and autistic traits, indicating trait-related modulation of IBS may diminish as social interaction becomes more dynamic and complex.

Collectively, the reviewed findings of IBS in individuals with ASD demonstrate that while IBS can be altered in individuals with ASD, its expression is highly sensitive to interpersonal dynamics, task context, and individual traits. While reduced IBS is often seen in demanding social tasks, it can be preserved or enhanced in supportive conditions, such as parent–child interaction or matched trait dyads. These findings suggest that IBS reflects dynamic social engagement. While promising, the inconsistencies across studies indicate that further research is needed to clarify its reliability and ecological relevance as a potential biomarker for autism-related social functioning.

**Table 2 brainsci-15-01113-t002:** Studies Examining IBS in Autism Spectrum Disorder (N = 11).

Dyad Type	Method	Authors	N Pairs	Brain Regions	Key Findings
Patient–Experimenter	fNIRS	Quiñones-Camacho et al. [27]	50	PFC, TPJ	Healthy controls exhibited significant IBS with the experimenter in bilateral TPJ during conversation, whereas adults with ASD showed reduced IBS in these regions.
					Lower IBS in the TPJ was associated with greater self-reported social communication impairment across the sample.
Parent–Child	fNIRS	Wang et al. [30]	16	Superior frontal cortex (SFC), mPFC, left lateral PFC (left-lPFC)	During cooperation, dyads exhibited greater IBS in both right and left SFC compared to solo conditions.
					However, greater autism symptom severity was associated with reduced IBS in the right SFC, particularly linked to deficits in communication and imagination.
Parent–Child	fNIRS	Kruppa et al. [26]	59	inferior frontopolar cortex (iFPC), bilateral dlPFC, mPFC, Brodmann area 8 (BA8), right frontopolar cortex (rFPC)	In typically developing dyads, IBS was observed in bilateral dlPFC and FPC during parent–child competition, and localized IBS in FPC during cooperation.
					IBS effects were modulated by child age, with competition-related IBS increasing and cooperation-related IBS decreasing with child age.
					In contrast, dyads with ASD showed no significant IBS increases compared to random pairs, and group comparisons revealed no robust differences.
Parent–Child	fNIRS	Minagawa et al. [29]	99	Bilateral temporal and frontal areas	During breastfeeding, dyads exhibited significantly stronger IBS compared to holding or control conditions. Synchrony involved the mother’s vPFC (esp. right OFC) and the infant’s TPJ (esp. left STG/STS), reflecting reward–motivation and early mentalization systems.
					Dyads with elevated likelihood of ASD showed stronger IBS in L-MTG–STG connections, potentially reflecting heightened maternal sensitivity.
Parent–Child	fNIRS	Tang et al. [28]	71	Fronto-parietal region	No significant group differences were found in IBS between ASD and TD dyads during parent–child cooperation. However, adopting a delayed response strategy significantly enhanced IBS in the right superior frontal cortex for both ASD and TD dyads, suggesting that cooperative strategy influenced degree of IBS.
					Parents of ASD children were more likely to adopt delayed strategies and adjusted their responses more strongly to align with their child’s behavior.
Child–Child	fNIRS	Zhou et al. [21]	23	Left frontal cortex, Right frontal cortex (rFC), Right temporo-parietal area (rTP), Left temporo-parietal area (lTP), fronto-temporo-parietal networks	Higher autistic traits were associated with smaller IBS increases during joint reading (lTP–rFC), while greater trait differences within dyads predicted stronger IBS during video watching without interactions (lTP–rFC). During Jenga play, trait differences showed no significant effect on IBS.
					Unique IBS connections predicted higher vs. lower autism spectrum quotient scores, showing distinct contributions of more autistic vs. less autistic children during interaction.
Adult–Adult	fNIRS	Peng et al. [20]	N = 64	bilateral PFC, temporo-parietal junctions	Dyads with high autistic traits (HH) exhibited enhanced IBS in the left STG and right frontopolar cortex compared to both LL and HL dyads during conversations.
					Despite atypical behavioral patterns (reduced mutual gaze, fewer conversational turns, less emotional sharing), HH dyads rated their conversations more positively and demonstrated effective communication through heightened neural synchrony.
Adult–Adult	fNIRS	Feng et al. [22]	N = 50	Right TPJ	Only LL dyads exhibited significant IBS in the rTPJ during conversations. HH dyads showed significantly lower IBS in the rTPJ compared to LL dyads, while HL dyads did not differ significantly from either group. Across all dyads, higher average autistic traits were associated with reduced IBS in the rTPJ, but IBS was not linked to trait differences between partners.
Adult– Child	fNIRS	Du et al. [31]	N = 76	Bilateral IFG, bilateral IPL, bilateral TPJ	In the right IPL, autistic children showed higher IBS during meaningful imitation but lower IBS during meaningless imitation compared to non-autistic peers. IBS in this region positively predicted autism for meaningful movements and negatively predicted autism for meaningless ones. In a follow-up study, non-autistic children again showed higher r-IPL IBS during meaningless imitation, with synchrony in transitive, gestural, and orofacial movements predicting lower autism likelihood.
Patient–Confederate	EEG	Key et al. [33]	N = 68	Temporo-parietal regions	Dyads with adolescents with autism exhibited greater IBS in bilateral temporoparietal regions during conversation compared to baseline. Increased IBS was associated with fewer autism symptoms (lower ADOS severity and social affect scores) and greater social competence (caregiver reports, theory of mind skills). Sex differences emerged: females with ASD showed significant IBS increases across theta, alpha, and beta bands, whereas males did not show significant condition differences.
Parent–Child	EEG	Kang et al. [32]	N = 7	Bilateral PFC, dlPFC, mPFC	Child–Parent dyads exhibited greater IBS in frontal regions compared to Child–Therapist dyads, with stronger synchrony during music sessions than storytelling. Directionality analyses showed stronger adult-to-child influence than child-to-adult influence.

### 3.2. Inter-Brain Synchrony and Stress

A total of 10 studies investigated IBS within stress contexts. Table 3 summarizes the studies related to IBS in stressful contexts. Stress was consistently found to modulate IBS across different types of dyads, whether between parents and children, spouses, or unrelated adults. Critical brain regions implicated in IBS under stressful conditions include the prefrontal cortex (PFC) and TPJ. These areas are integral to higher-order cognitive and social processes such as emotional regulation, social cognition, and theory of mind. The PFC, particularly the dlPFC, plays a vital role in executive functions, cognitive empathy, and stress regulation. Similarly, the TPJ is central in distinguishing between self-related and other-related mental states, supporting social understanding and cooperation.

However, the direction of these effects varied. Several studies found that stress related to parenting and socioeconomic hardship was associated with reduced IBS within parent–child dyads [34,35,36]. In contrast, other studies found that IBS increased when dyads were placed under stress when working on a collaborative problem-solving activity [37,38]. Notably, some studies found that IBS was modulated by successful collaborative strategies [39], extended parent–child play [40], and children’s behaviors during parent–child active interactions [35,41].

#### 3.2.1. Inter-Brain Synchrony and Stress in Parent–Child Dyads

Several studies have specifically focused on parent–child brain synchrony and parenting stress. Parent–child IBS reflects coordinated neural activity between the brains of parent and child during joint interactions, demonstrating essential implications for the child’s socio-emotional development and overall attachment quality [42].

Recent studies utilizing fNIRS have demonstrated how parenting stress distinctly modulates IBS, influencing the quality and effectiveness of parent–child interactions. Elevated parenting stress is linked to reduced mother-child IBS, particularly in prefrontal regions associated with social cognition, mentalizing, and emotion regulation, during passive interactions such as co-viewing animations [34]. These regions, the IFG, dlPFC, and frontal eye fields, are known to support cognitive empathy and theory of mind processes, highlighting how parental stress may disrupt essential social-cognitive processes during everyday interactions.

In more active, naturalistic contexts, higher neural synchrony between mothers and children in the bilateral PFC and temporo-parietal areas was associated with more reciprocal interactions and predicted better cooperative problem-solving performance [35]. This study demonstrated that transient, state-related factors such as maternal stress attenuated neural synchrony, while trait-like factors like child temperament did not exert a significant effect. Notably, higher active participation from the child was linked to enhanced IBS, suggesting that the child’s behavioral involvement may support stronger neural alignment. These findings suggest that maternal stress can impair neural attunement during real-time interactions, whereas active child participation may buffer against such disruptions, reinforcing the dynamic and situation-dependent nature of parent–child neural synchrony.

While stress often attenuates IBS, other evidence suggests that engaging in prolonged positive joint activities, such as shared play, may serve as a compensatory mechanism enhancing IBS in mother-child dyads with elevated parenting stress. Azhari [40] examined parent–child IBS during active play and found that dyads with higher parenting stress exhibited greater IBS in the left PFC, but only after a sustained 35 s period of uninterrupted positive joint engagement. This suggests that, for parents who are experiencing elevated stress, prolonged moments of shared positive behavior can entrain neural synchrony and potentially offer reparative social benefits.

Using a subset of participants from [40], Liu [41] extended the analysis to both passive video co-viewing and active play conditions. Higher neural synchrony was observed in mother–child dyads compared to father-child dyads during shared video viewing and self-reported parenting stress attenuated IBS across both dyad types. However, during active play, no significant overall differences in IBS were found between mother– and father–child dyads, and stress no longer significantly modulated IBS. Instead, moment-specific child behaviors shaped neural coupling: mother–child IBS increased in the left frontal regions, especially in the superior frontal gyrus (SFG), middle frontal gyrus (MFG) and IFG, when the child smiled, aligning with the role of the SFG in emotional expression. On the other hand, father–child IBS in the right MFG, right IFG and bilateral SFG increased when the child spoke, recruiting the regions involved in attention and action regulation. These patterns support the notion that mothers and fathers engage differently with their children, with mothers more attuned to low-arousal emotional cues, and fathers more responsive to high-arousal, cognitively demanding interactions.

Together, these findings suggest that while parenting stress can suppress IBS during passive interactions, extended interactions (such as joint play) and child’s reciprocal behaviors may enhance IBS in parent–child dyads. Notably, the differing patterns observed in [41] indicate that the mechanisms underlying this IBS enhancement may vary by parent gender. Understanding these gender-specific neural mechanisms can provide insights into tailored interventions aimed at enhancing parent–child neural synchrony and child developmental outcomes under stress.

#### 3.2.2. Inter-Brain Synchrony and Stress in Induced Stress Contexts

Expanding beyond parent–child dyads, recent research employed an induced stress paradigm to investigate IBS patterns during acute stress and how it may relate to cooperative outcomes and communication. Several studies observed that under induced acute stress, adult-adult dyads exhibited greater IBS in the prefrontal areas, such as dorsolateral and frontopolar cortices and the TPJ [37,38,43,44], particularly the rTPJ. These studies reported a significant association between stress and IBS without establishing causality. For example, two studies [38,43] observed increased IBS in the rTPJ following stress induction during cooperative tasks in female participants. Similarly, Zhang [44] reported elevated IBS in the right dlPFC under stress, alongside competitive behaviors. Building on these correlational findings, Zhao [37] provided compelling evidence for causal relationships using Granger causality analyses. Their study showed that acute stress enhanced cooperative performance by increasing brain-to-brain coherence in the rTPJ, with this neural synchrony and self-other overlap serially mediating the effect of stress on cooperation. This work underscores the mechanistic role of IBS in translating stress effects into improved social task performance.

Extending beyond romantic and cooperative dyads, Hoyniak [36] examined parent–child pairs under experimentally induced frustration and found that higher sociodemographic risk, rather than familial risk factors like household chaos, was associated with reduced IBS in the lateral PFC. Interestingly, neural synchrony normalized during recovery, suggesting that acute stress disrupts parent–child brain coupling primarily under challenging environmental conditions. These findings complement earlier work such as [39] which linked naturally occurring parenting stress, rather than induced stress, with increased IBS during cooperative strategies in married parents.

Together, these studies collectively demonstrate that stress, whether experimentally induced or naturally occurring, modulates IBS across different social contexts. Importantly, causal evidence suggests that IBS not only responds to stress but also mediates its effects on collaborative performance, highlighting neural synchrony as a key mechanism linking stress to social interaction outcomes.

**Table 3 brainsci-15-01113-t003:** Studies Examining IBS in Stress Contexts (N = 10).

Dyad Type	Method	Authors	N Pairs	Brain Regions	Key Findings
Patient–Patient	fNIRS	Zhang et al. [44]	N = 40	Right dlPFC, FPC, left middle PFC, right OFC	Acute psychosocial stress promoted cooperation, accompanied by increased IBS in the right FPC, right OFC, left middle PFC. Stress also facilitated competition, with significantly greater IBS in the right dlPFC, which correlated positively with competition intensity.
Parent–Child	fNIRS	Azhari et al. [34]	N = 31	SFG, MFG, IFG, anterior PFC (aPFC), dfPFC	Dyads with higher parenting stress showed lower IBS in the medial left PFC (including IFG, FEF, dlPFC).
Parent–Child	fNIRS	Nguyen et al. [35]	N = 42	bilateral PFC and temporo-parietal areas	During cooperation, dyads exhibited higher IBS in both the bilateral PFC and TPJ compared to individual or resting conditions. Higher behavioral reciprocity was linked to stronger IBS in these regions, which in turn predicted better problem-solving performance. Conversely, maternal stress was associated with reduced IBS across regions of interest, while greater child agency corresponding to enhanced IBS during cooperation.
Parent–Child	fNIRS	Hoyniak et al. [36]	N = 115	dlPFC	Higher sociodemographic risk was associated with reduced parent–child IBS in the lateral PFC during experimentally induced frustration. Familial risk (household chaos, family conflict, caregiver psychopathology) was not significantly related to IBS. Neural synchrony returned to typical levels during recovery, suggesting adversity mainly disrupts synchrony under acute stress.
Parent–Child	fNIRS	Azhari et al. [40]	N = 60	frontal left and posterior right areas of the PFC	During joint play, dyads with higher parenting stress exhibited greater IBS in the frontal left PFC, but only after sustained interaction (35 s window). Parenting stress was also associated with decreased IBS in the posterior right PFC. Mother–child dyads showed significantly greater IBS than father–child dyads in this region.
Parent–Child	fNIRS	Liu et al. [41]	N = 62	Bilateral SFG, bilateral MFG, bilateral IFG, and aPFC	During video co-viewing, mother–child dyads exhibited stronger IBS in aPFC and bilateral IFG compared to father–child dyads, particularly when videos were visually complex.
					IBS during co-viewing was negatively associated with parental distress, a subscale of parenting stress.
					During free play, mother–child dyads showed stronger IBS in left frontal regions (SFG, MFG, IFG) associated with child involvement, while father–child dyads showed stronger IBS in right frontal regions (SFG, MFG, IFG) during child verbalizations.
					IBS during active play was negatively associated with child difficulty reported by parents.
Spousal partners	fNIRS	Tang et al. [39]	N = 43	right frontal-parietal regions and right frontal cortex	During cooperative tasks, couples adopting a delayed response strategy exhibited stronger IBS in the right frontal cortex compared to those using immediate or no clear strategy.
					Greater parenting stress was associated with a higher likelihood of adopting the delayed response strategy, which in turn predicted stronger IBS in the right frontal cortex.
Adult–Adult	fNIRS	Zhao et al. [43]	N = 84	Frontal eye fields, dlPFC, Middle frontopolar area, Inferior frontopolar area, Primary somatosensory cortex, Supramarginal gyrus part of Wernicke’s area, Subcentral area, Supramarginal gyrus part of Wernicke’s area, STG	Under acute stress, dyads exhibited greater IBS in the frontopolar cortex and left TPJ during late decision-making rounds compared to control dyads. Increased IBS at the TPJ was positively correlated with higher cooperativeness and more rational group decision-making performance, supporting the “tend-and-befriend” stress response pattern.
Adult–Adult	fNIRS	Zhao et al. [37]	N = 44	Right fronto-tempo-parietal region	Under acute stress, dyads exhibited greater IBS in the right TPJ during cooperation compared to control dyads.
					Increased IBS in this region was correlated with higher self-other overlap and better cooperative performance, with mediation analyses suggesting that IBS and self-other overlap serially mediated the effect of stress on cooperation.
Adult–Adult	fNIRS	Lin et al. [38]	N = 40	rTPJ	Under acute stress, dyads exhibited greater IBS in the rTPJ compared to control dyads during the Joint Simon Task.
					This enhanced synchrony at the rTPJ was interpreted as a neural mechanism supporting self–other distinction and shared intentionality under stress.

### 3.3. Inter-Brain Synchrony Within the Contexts of Anxiety and Depression

A total of five studies investigated anxiety and IBS, with two of these also assessing depressive symptoms [45,46]. Table 4 summarizes the studies related to IBS with relation to anxiety and depression. Most studies examining inter-brain synchrony in the context of anxiety and depression focus on emotional processing and regulation, highlighting how neural alignment between individuals supports or is disrupted during affective experiences and attempts at emotional control.

Among these studies, EEG hyperscanning was the most frequently employed. This technique offers exceptional temporal resolution, enables real-time monitoring of neural activity between interacting individuals. This approach is especially valuable in capturing transient emotional and social dynamics during face-to-face interactions, such as therapeutic settings or social anxiety-provoking scenarios [47,48]. In contrast, fNIRS, despite its lower temporal resolution, provides superior spatial resolution, making it suitable for monitoring brain regions involved in verbal communication and socio-emotional processing in more dynamic or ecologically valid settings [45,49]. Hyperscanning using fMRI complements these approaches by offering extensive spatial coverage and allowing the observation of synchronized neural activation across broader brain networks in naturalistic settings, further enriching our understanding of neural synchrony in relation to anxiety and depression [46].

#### Internalizing Symptoms and Inter-Brain Synchrony in Dyadic Interactions

Anxiety significantly modulates inter-brain synchrony across diverse social dyadic interactions, including parent–child and peer relationships. In adolescent–parent interactions, heightened social anxiety manifests in differential gamma-band synchrony during emotional stimuli, suggesting distinct neural mechanisms underlying emotional regulation within anxious dyads [48]. Adolescent-parent dyads with low social anxiety showed heightened gamma interbrain synchronization in the central and parietal brain regions during negative emotional conditions compared to neutral conditions. Interestingly, however, dyads with higher social anxiety were associated with increased gamma interbrain synchronization in the parietal regions during positive emotional conditions when compared to negative conditions. Additionally, within positive conditions, dyads with higher social anxiety showed stronger IBS in parietal regions compared to central regions, highlighting distinct neural patterns associated with social anxiety across different emotional contexts.

Beyond moment-to-moment emotional regulation in parent–child interactions, recent research has further clarified how negative family emotional climates relate to internalizing symptoms through altered neural synchrony in parent–child dyads. Su [46] found that children from families with a negative emotional climate showed more severe internalizing symptoms (e.g., anxiety and depression) and externalizing symptoms (e.g., aggression and rule-breaking). Reduced inter-brain synchrony between the vmPFC and the hippocampus in parent–child dyads during movie watching was linked to these family environments. This reduced connectivity specifically mediated the link between negative emotional climate and children’s internalizing symptoms. These findings suggest that atypical IBS within parent–child interactions represents a neurobiological pathway through which adverse familial emotional environments influence anxiety and depressive symptoms in children.

Therapeutic contexts also offer another important domain for studying IBS in relation to anxiety and depression. Frontal alpha asymmetry (FAA), prominently assessed via EEG, has emerged as a neural indicator of emotional processing and motivational states associated with anxiety. In a dual-EEG study of Guided Imagery and Music therapy [47], temporal alignment of frontal alpha asymmetry between therapist and client was observed during emotionally significant moments, indicating synchronized neural processing of anxiety and negative emotions. These moments, involving difficult themes such as loss and messages of hope, showed dynamic shifts in emotional valence reflected by fluctuations in frontal alpha asymmetry. These fluctuations in FAA corresponding with key emotional turning points and breakthroughs in emotional regulation suggest the potential for IBS in FAA signals to indicate real-time emotional adjustments. The significant cross-correlation between the Guide’s and Traveler’s neural activity suggests partial inter-brain synchrony, highlighting mutual emotional regulation and attunement in the therapeutic dyad. This study demonstrates how IBS can capture the moment-to-moment neural coupling underlying shared emotional experiences and anxiety management in therapy.

Behavioral findings indicate that interpersonal emotional regulation techniques, particularly active listening, effectively reduce anxiety and enhance emotional regulation during social interactions. Wang [45] reported increased activation in the OFC, bilateral dlPFC, and rTPJ during active listening. Additionally, IBS increased in these same regions across different frequency bands. However, the increases in IBS were not significantly correlated with participants’ levels of depression, anxiety, or empathy. Specifically, the increased activation and synchronization in these neural areas likely facilitate improved emotional attunement, empathy, and cognitive appraisal, which are central to successful emotional regulation in social settings. These enhancements in neural synchrony underscore the importance of active listening as a potential neuroscience-backed intervention for anxiety management in interpersonal contexts.

Finally, the broader social context markedly influences inter-brain synchrony and associated anxiety responses. Anxiety induced by socioeconomic disparity significantly modulates interpersonal neural synchronization, especially in regions associated with social cognition and emotional regulation [49]. For example, dyads characterized by higher socioeconomic disparities demonstrate increased synchronization within critical neural regions such as the dlPFC, FPC, and pars triangularis [49]. This heightened synchronization likely represents greater cognitive and emotional efforts to manage anxiety stemming from perceived social differences, stereotypical biases, and the stress associated with navigating interactions across socioeconomic boundaries. These neural adjustments reflect active attempts to regulate prejudices and enhance communication, underscoring the role of the frontal cortical regions in mediating socio-emotional processes during socially demanding interactions.

Altogether, the number of studies investigating IBS in anxiety and depression remains limited and varies across contexts and factors. The existing research highlights consistent patterns of altered IBS in key regions involved in emotion regulation and social cognition. These findings suggest that IBS may offer valuable insights into neural mechanisms underlying socioemotional deficits commonly associated with these conditions. However, further research targeted at clinical populations is needed to establish its potential as a biomarker for anxiety or depression. Continued exploration into these dynamics holds promise for refining interpersonal and family-centered therapeutic interventions targeting emotional regulation and depressive disorders.

**Table 4 brainsci-15-01113-t004:** Studies Examining IBS in Anxiety and Depression Contexts (N = 5).

Dyad Type	Device	Authors	N Pairs	Brain Regions	Key Findings
Adult–Adult	fNIRS	Descorbeth et al. [49]	N = 39	left dlPFC (BA46), frontopolar area, and Broca’s Area ROI (pars triangularis, BA45), Wernicke’s Area ROI (supramarginal gyrus, BA40; angular gyrus, BA39; and STG, BA22)	During live prosocial dialogue, dyads with higher socioeconomic disparity exhibited greater IBS in left dlPFC, frontopolar area, and pars triangularis compared to low-disparity dyads. Cross-brain coherence analyses further showed stronger coupling between pars triangularis and pre-/supplementary motor cortex for high-disparity dyads.
Adult–Adult	fNIRS	Wang et al. [45]	N = 20	OFC, bilateral dlPFC, rTPJ	During active listening, dyads exhibited significant IBS increases in the medial PFC, OFC, bilateral dlPFC, and right TPJ compared to baseline. IBS increments in these regions provided neural evidence for the role of active listening in interpersonal emotion regulation.
					No significant associations were found between IBS and depression, anxiety, or empathy scores of the emotion regulator.
Patient–Clinician	EEG	Fachner et al. [47]	N = 1	Bilateral frontal regions of the brain, Parietal regions and Left middle temporal gyrus	Higher IBS during “moments of interest” in therapy was found compared to control moments. Synchrony particularly pronounced in rTPJ and frontal regions.
Parent–Child	EEG	Deng et al. [48]	N = 25	Central and parietal lobe	Adolescent social anxiety modulated interbrain synchrony during emotional processing. Dyads with high social anxiety adolescents exhibited greater gamma IBS at parietal sites when processing positive stimuli, but reduced IBS at parietal and central sites when processing negative stimuli compared to low social anxiety dyads.
Parent–Child	fMRI	Su et al. [46]	N = 395	Dorsal mPFC (dmPFC), vmPFC, hippocampus, IFG, middle cingulum gyrus, precuneus, fusiform, hippocampus, middle occipital gyrus, amygdala	Compared to child–stranger dyads, child–parent dyads exhibited greater IBS in the dmPFC and vmPFC and stronger vmPFC connectivity with socioemotional regions (hippocampus, amygdala, precuneus, fusiform).
					Negative family emotional climate was associated with reduced child–parent vmPFC–hippocampal connectivity, which mediated higher child internalizing symptoms, particularly anxious/depressed symptoms.

### 3.4. Inter-Brain Synchrony and Other Disorders: Addiction, Personality Disorders, Aggression and Psychosis

A total of five studies investigated IBS in the following contexts: clinical high risk for psychosis, alcohol use disorder (AUD), gaming disorder (GD), and borderline personality disorder (BPD). These studies employed either fNIRS or fMRI hyperscanning techniques. Table 5 shows the studies examining IBS in other social disorders.

#### 3.4.1. Inter-Brain Synchrony and Alcohol Use Disorder (AUD)

Alcohol use disorder (AUD) significantly impacts social cognition and neural coordination during interpersonal interaction. Guo [50] utilized fNIRS hyperscanning to compare dyads of individuals with AUD (N = 14 dyads) to dyads of healthy controls (N = 18 dyads). During both cooperation and competition tasks, individuals with AUD demonstrated significantly decreased IBS in the right middle frontal cortex. This region, which includes components of the dlPFC, is vital for social attention, regulation of social feedback, and cognitive control during interactions. The reduced IBS was negatively correlated with non-planning impulsivity, suggesting that impaired self-regulation may underlie disrupted social synchronization in AUD [50]. These findings align with prior literature showing that the PFC in AUD individuals is structurally and functionally compromised, contributing to emotional dysregulation and impaired social cue processing. The results imply that reduced neural coupling in key frontal regions may be a mechanism of social dysfunction in AUD and potentially predictive of treatment outcomes and relapse.

#### 3.4.2. Inter-Brain Synchrony, Gaming Disorder (GD) and Hazardous Gaming (HG)

Gaming disorders are associated with social withdrawal and impaired interpersonal functioning. Huang [51] explored IBS among adolescents and young adults with GD or HG behaviors during cooperative video gameplay. Using fNIRS hyperscanning, the study identified decreased IBS in the left dlPFC in dyads involving one or both individuals with GD or HG. This reduction in synchrony was significantly correlated with peer relationship problems, indicating that disrupted interpersonal neural coordination mirrors the social difficulties seen in gaming-addicted populations. The dlPFC has been associated with cooperative behavior, self-regulation, and theory of mind. The impaired synchrony in this region implies that individuals with GD or HG may struggle with joint attention and emotional attunement during real-world cooperation. These insights could inform behavioral therapies targeting social cognition deficits in internet gaming addiction.

#### 3.4.3. Inter-Brain Synchrony and Borderline Personality Disorder (BPD)

BPD is characterized by chronic interpersonal instability and emotional dysregulation. Bilek [52] employed a dual-fMRI hyperscanning design to investigate neural coupling in dyads involving individuals with current BPD (cBPD), remitted BPD (rBPD), and healthy controls (HC). During a joint attention task, cBPD-HC pairs exhibited significantly lower neural coupling in the rTPJ, a region central to mental state attribution and social inference. Interestingly, rBPD-HC dyads showed IBS levels similar to pairs of HC-HC pairs, suggesting that neural markers of social interaction impairment in BPD may be state-dependent and potentially reversible with treatment. Furthermore, the degree of neural coupling was inversely associated with self-reported childhood adversity, indicating that early life trauma may disrupt neural systems underlying dyadic synchrony. These findings underscore the relevance of IBS as a potential indicator of relational dysfunction in BPD, although more research needs to be conducted to assess the replicability of findings.

#### 3.4.4. Inter-Brain Synchrony and Psychosis

In a study by Wei [53], IBS was assessed in individuals at clinical high risk (CHR) for psychosis. The study compared CHR–healthy control (CHR–HC) dyads with HC–HC pairs during cooperative and competitive tasks using fNIRS hyperscanning. Results revealed that during cooperation, CHR-HC dyads exhibited significantly reduced IBS in the right IFG (rIFG). Moreover, reduced IBS was correlated with increased symptom severity in domains such as suspiciousness and motor disturbance. The rIFG is associated with inhibitory control, empathy, and social cognition. Reduced IBS in this region may signify early neurobiological disruption in social interaction systems, offering a potential marker for identifying individuals at risk of conversion to psychosis. Taken together, emerging IBS findings in this area reveal disrupted IBS patterns in the right IFG, right middle frontal cortex, and dlPFC in these clinical populations. These findings indicate that IBS is sensitive to socio-cognitive and emotional impairments across psychiatric conditions, although the robustness of its use as a transdiagnostic marker needs to be verified with future research.

**Table 5 brainsci-15-01113-t005:** Studies Examining IBS in Other Disorders (N = 5 studies).

Condition	Dyad Type	Device	Authors	Sample Size (in Pairs)	Brain Regions	Key Findings
Clinical High Risk (CHR) of Psychosis	Patient–Healthy Controls	fNIRS	Wei et al. [53]	N = 38	Right IFG	During cooperation, dyads consisting of a member with CHR of psychosis exhibited reduced IBS in the right IFG compared to healthy control dyads.
						Reduced IBS was positively correlated with more severe suspiciousness/persecutory ideas and movement disorder symptoms, and with lower cooperation performance.
Alcohol Use Disorder (AUD)	Patient–Healthy Controls		Guo et al. [50]	N = 42	Right middle frontal cortex	Dyads with AUD exhibited reduced IBS in the right middle frontal cortex during both cooperation and competition compared to healthy controls.
						Within AUD dyads, lower IBS during cooperation was significantly associated with greater non-planning impulsivity.
Gaming Disorder (GD)	Parent–Child		Huang et al. [51]	N = 34	PFC, left ventrolateral PFC (vlPFC), left dlPFC	Dyads including individuals with GD or hazardous gaming (HG) behaviors exhibited reduced IBS in the left dlPFC compared to HC–HC dyads during cooperation.
						A marginal reduction in IBS was also observed in the right vlPFC across groups.
						Within GD/HG dyads, higher IBS in the left dlPFC correlated with greater self-reported peer problems, suggesting disrupted social alignment processes in gaming disorder.
Aggressive rule-breaking behaviors	Parent–Child	fMRI	Su et al. [46]	N = 395	dmPFC, vmPFC, hippocampus, IFG, middle cingulum gyrus, precuneus, fusiform, hippocampus, middle occipital gyrus, amygdala	Stronger vmPFC–emotion system synchrony emerged in parent–child dyads during emotional movie scenes. While reduced synchrony was linked to higher internalizing symptoms, it was not linked to higher externalizing symptoms, such as aggressive and rule breaking behaviors, in children within negative family environments.
Borderline Personality Disorder	Patient–Healthy Control		Bilek et al. [52]	N = 60	rTPJ	BPD-Healthy control dyads showed significantly reduced IBS in the rTPJ than HC–HC dyads.

### 3.5. Consistent Neural Hubs Across Disorders

Figure 4 presents a heatmap illustrating the brain regions implicated across different conditions, with green indicating involvement and red indicating absence. Convergent activation is observed in the prefrontal and temporoparietal regions, particularly the dlPFC, mPFC, vmPFC, TPJ, IFG, and OFC, which appear consistently across all disorders. In contrast, limbic structures such as the amygdala and hippocampus show more selective involvement, primarily in anxiety and depression.

## 4. Discussion

The present systematic review sought to address two fundamental research questions concerning the role of IBS in psychological and neurodevelopmental conditions: (1) whether IBS can reflect underlying psychopathologies and potentially serve as a neurophysiological biomarker, and (2) which specific brain regions are consistently implicated across different clinical populations. Drawing from recent hyperscanning studies utilizing fNIRS, EEG, and fMRI technologies, this review reveals compelling evidence that IBS is a sensitive, dynamic, and contextually modulated marker of interpersonal emotional and cognitive processes—one that is significantly altered in various psychiatric conditions. The findings underscore the immense clinical and theoretical potential of IBS as a diagnostic and interventional tool.

### 4.1. IBS as a Neurophysiological Biomarker of Psychological Disorders

The evidence reviewed provides preliminary support for the first research question: generally, it appears that reduced IBS potentially reflects underlying psychological conditions. Across disorders such as ASD, stress, anxiety, and depression, atypical IBS patterns have been repeatedly observed, suggesting that modulations in neural synchrony may serve as indicators of emotional dysregulation and impaired social functioning.

Across different dyad pairings and settings, reduced IBS has been observed during socially and emotionally demanding tasks, such as regulating negative emotions or behavioral imitation. However, our review has also uncovered that dynamic modulation in IBS patterns depends on state-like factors such as parental emotional states, child behaviors, presence of similar traits in partners, type of emotion regulation strategy as well as environmental factors such as family socio-emotional climate. Collectively, these findings support the conceptualization of IBS as both a trait-like indicator of social-emotional vulnerabilities and a state-sensitive measure of contextual modulation, offering unique diagnostic and prognostic advantages in clinical psychology.

A central challenge in interpreting IBS as a potential biomarker for psychological conditions lies in the absence of a standardized baseline. Establishing whether IBS is “reduced” in a given condition presupposes a reference point, yet no consensus currently exists regarding what constitutes a normative or gold-standard level of IBS. This issue is compounded by substantial variability in study designs across the literature, including differences in comparison dyads (e.g., patient–control, matched peers, confederates), interaction contexts (e.g., cooperative vs. passive tasks), and task types (e.g., cognitive, emotional, naturalistic) as observed in our reviewed studies here. Such heterogeneity makes it difficult to determine whether observed differences in IBS are attributable to psychopathological features, task demands, or contextual factors. As a result, claims about aberrant IBS patterns should be interpreted with caution, and future research would benefit from greater standardization in experimental paradigms or the development of normative IBS datasets to serve as reference points.

Taken together, these findings suggest that IBS can reflect socio-emotional vulnerabilities and state-dependent adaptations across psychological disorders, thereby offering strong potential as a biomarker. However, it must be emphasized that the current evidence is correlational and heterogeneous, and few direct causal associations with specific psychopathologies can yet be established. As with many psychiatric conditions where clinical indicators remain underdeveloped, the literature is still evolving. At present, IBS should not be regarded as a clinical indicator but rather as a promising neurophysiological index of interpersonal adaptation and vulnerability that holds potential to inform future biomarker development.

### 4.2. Consistent Neural Hubs Across Disorders: The Social Brain Network

In this review, we also identified a core set of brain regions repeatedly implicated in IBS across clinical conditions. These include the dlPFC, mPFC, vmPFC, TPJ, IFG, and OFC. The dlPFC is a core region involved in executive functions such as working memory, cognitive control, and the regulation of emotional responses [54,55,56]. Its consistent presence across disorders suggests that IBS in the dlPFC may support shared cognitive regulatory processes during social interaction, and that dysfunction in this region may reflect compromised top-down regulation in clinical populations.

The TPJ, also consistently implicated, is a key node in the social brain network. It plays a central role in perspective-taking, theory of mind, and the attribution of mental states to others [57,58]. IBS in this region likely reflects shared attentional and mentalizing processes between dyad members, which are particularly important in contexts requiring empathy, joint attention, and mutual understanding. These processes are often disrupted in conditions such as autism and anxiety.

The mPFC and vmPFC are central to self-referential processing and affective meaning-making [59,60,61] and are especially salient in familial and caregiver-child interactions [62,63]. Given the mPFC’s role in self-other processing, reflective thinking, and integrating emotional and social information, its consistent appearance suggests that these higher-order, relational cognitive processes are commonly disrupted across different mental health contexts.

Limbic structures, including the amygdala and hippocampus, appeared more selectively, with involvement primarily in studies of anxiety and depression. The amygdala is critical for detecting emotional salience and threat [64], while the hippocampus contributes to memory consolidation and contextual emotional processing [65]. Their selective involvement in IBS during these conditions may reflect heightened emotional reactivity and internalizing symptomatology and suggest that emotional memory and salience may become synchronized during interpersonal interactions in the context of shared distress. In contrast, these limbic areas are not consistently implicated in studies of autism or stress, possibly due to different patterns of emotional engagement or regulatory demands. For instance, autism-related IBS findings tend to emphasize social-cognitive regions like the TPJ over limbic circuits, consistent with the social communication challenges typical of the disorder. Finally, the OFC facilitates the processing of rewards and social feedback [66]. Studies on anxiety and depression highlight increased IBS in this region during interventions like active listening, which promote interpersonal regulation and connection. These findings reinforce the idea of a core “social brain” network, wherein inter-brain synchronization supports mutual understanding, shared emotions, and collaborative decision-making.

Altogether, these findings highlight both convergence and divergence in the neural architecture of IBS across clinical conditions. While the prefrontal cortices and TPJ may represent transdiagnostic hubs for interpersonal alignment and regulation, variability in the involvement of regions like the amygdala and hippocampus point to condition-specific profiles of neural synchrony. These patterns contribute to our understanding of how distinct psychopathologies may uniquely alter shared neural dynamics and suggest potential pathways through which IBS might serve as a biomarker for interpersonal dysfunction in mental health.

### 4.3. Contextual and Dyadic Modulation of IBS

A particularly striking conclusion is that IBS is highly modulated by context, including dyad type, emotional state, gender, and interaction strategy. Parent–child, therapist–patient, romantic partner, and peer dyads exhibit different IBS patterns depending on emotional proximity, stress, and relational history [18,20,21,33,40,47]. Gender differences are also evident. Mothers consistently show higher synchrony than fathers during shared tasks with children, especially in prefrontal regions [40,41]. This may reflect both biological attunement mechanisms and socialized caregiving practices that prioritize emotional engagement. Moreover, strategic behaviors have been shown to enhance IBS. For example, delayed response strategies during cooperative problem-solving increase IBS and lead to more efficient outcomes [39]. Similarly, prolonged play or active listening enhances IBS, offering evidence for the neural benefits of relational attunement and co-regulation [40,45]. These findings emphasize the potential for targeted behavioral interventions to enhance IBS in clinical contexts.

### 4.4. Implications for Clinical Practice and Future Research

The findings of this review have significant implications for clinical practice, particularly in the development and evaluation of interventions targeting interpersonal functioning. IBS emerges as a promising index that can objectively track changes in social and emotional attunement; these facets that often elude behavioral observation alone. In therapeutic contexts, IBS could be utilized to monitor treatment progress in real time, capturing subtle improvements in relational dynamics and emotional resonance. For example, dyadic interventions such as parent–child play therapy, music-based therapies, or guided parent coaching have shown potential in enhancing IBS and could be further optimized using IBS as a neurofeedback tool [24,40]. Moreover, IBS may play a pivotal role in early detection and prevention. Reduced synchrony in parent–child dyads or among individuals at CHR for psychosis has been linked to elevated emotional and behavioral vulnerabilities [46,53]. As hyperscanning technologies continue to evolve toward greater portability and ecological validity, longitudinal applications of IBS could allow clinicians to assess developmental trajectories and risk profiles across time and contexts.

It is important to note that IBS itself is not conceptualized as a direct cause of these disorders; rather, IBS represents a neurophysiological correlate or marker of interpersonal functioning that is often altered by these psychological conditions. Specifically, disruptions in IBS among individuals experiencing anxiety or depression likely reflect underlying impairments in social cognition, emotional attunement, or emotion regulation. Anxiety, for example, can hinder an individual’s ability to effectively engage in social interactions due to heightened emotional arousal, attentional biases, or fear of negative evaluation, thereby disrupting real-time neural synchronization [48,49]. Similarly, depression is frequently linked with emotional disengagement and reduced empathy, significantly limiting the potential for interpersonal neural synchrony during interactions [46].

To date, studies on IBS have focused primarily on the neural network level, with little investigation conducted on molecular processes that support these systemic observations. For instance, the specific brain regions frequently highlighted in IBS studies, such as the dlPFC, TPJ, and mPFC have well-established roles in synaptic signaling processes underlying social cognition and emotional regulation [67]. However, almost all current IBS studies have examined brain activity at the systemic level, without direct assessment of molecular synaptic signaling. Future research involving cellular-level examination of synaptic signaling within these critical regions could significantly enhance our understanding of the precise mechanisms underlying IBS, thereby strengthening the potential of IBS as a robust biomarker for psychological and neurodevelopmental conditions.

Although our review primarily concentrated on neuroimaging studies utilizing hyperscanning techniques, molecular players within the nervous system, namely neurotransmitters and neuropeptides potentially contribute to IBS [68]. Neurotransmitters such as serotonin and dopamine, well-documented in their roles in mood disorders, are likely to indirectly affect IBS through their modulation of social motivation, emotional responsiveness, and cognitive biases. Dysregulation of serotonin, frequently observed in anxiety and depression, can impair social and emotional processing, thus indirectly reducing IBS [69]. Dopaminergic systems, critical in reward processing and social motivation, may similarly impact neural synchronization by modulating an individual’s motivation to engage socially, particularly under conditions associated with depressive symptoms.

Additionally, neuropeptides such as oxytocin significantly influence social bonding, emotional closeness, and trust—processes integral to IBS. Elevated oxytocin levels are consistently linked to enhanced prosocial behavior, emotional attunement, and empathy, thereby facilitating increased neural synchrony during interpersonal exchanges. Conversely, disruptions in oxytocin signaling, commonly reported in autism and mood disorders, may impair social attunement and subsequently reduce IBS [70]. Furthermore, cortisol, a critical stress hormone, is known to affect cognitive control and emotion regulation, indirectly modulating IBS by influencing social interaction quality during acute stress [69,70,71]. Future research involving cellular-level examination of synaptic mechanisms, such as neurotransmitter release, receptor activation, and intracellular signaling cascades, within IBS-relevant regions could substantially refine our understanding of the mechanisms driving IBS. This line of inquiry would enhance the translational potential of IBS as a biomarker for psychological and neurodevelopmental conditions.

### 4.5. Limitations

Despite the strengths of this systematic review, several limitations should be acknowledged. First, we did not include a formal risk of bias (RoB) table, as there is currently no standardized appraisal tool specifically designed for hyperscanning studies. Existing tools, such as Cochrane RoB 2.0 or ROBINS-I, are intended for randomized or observational trials and do not readily apply to neuroimaging-based, dyadic research contexts (e.g., EEG, fNIRS, fMRI). Given the substantial variation in task designs, analytic pipelines, participant groups, and imaging protocols across studies, applying a single unified RoB tool would likely introduce more confusion than clarity [72,73].

Technical limitations of hyperscanning itself also warrant caution. Differences in preprocessing steps, motion correction strategies, and the constraints of specific modalities—such as fNIRS’s limited depth and spatial resolution—can affect data quality and interpretation [8]. These challenges are often compounded in interpersonal setups, where movement artifacts are more difficult to isolate. Moreover, analytic approaches to IBS vary widely across studies, with some using wavelet coherence, others using phase-locking values or Granger causality, making it difficult to directly compare results [8]. Another issue is the inconsistency in reporting study outcomes. Many studies did not report effect sizes, and when they did, the metrics used varied widely, ranging from Cohen’s d to partial η^2^ and correlation coefficients. Additionally, due to the unbalanced distribution of studies across conditions, frequency-based or hierarchical clustering analyses would not have been quantitatively intuitive. Such approaches could yield misleading groupings driven by unequal study representation rather than genuine neural convergence. Due to this lack of standardization and transparency, conducting a meaningful meta-analysis was not feasible for this review, hindering the quantitative synthesis of this paper. Future meta-analytic work incorporating standardized, weighted, or continuous frequency data may enable more robust quantitative synthesis and facilitate the use of clustering analyses to identify patterns of regional convergence across conditions.

A related challenge is the complexity of interpreting IBS itself. Most studies in this review employed cross-sectional designs, which limits our ability to comment on the temporal stability or predictive value of IBS as a clinical biomarker. Longitudinal evidence remains scarce, and without it, it is difficult to determine whether IBS differences reflect stable traits or state-dependent fluctuations.

Moving forward, it will be important for the field to adopt more standardized task protocols and preprocessing pipelines to improve replicability. Longitudinal studies are particularly needed to assess the stability and clinical utility of IBS over time. Additionally, consistent reporting of effect sizes and methodological details—including motion correction and artifact handling—will be critical for enabling future meta-analytic efforts and strengthening the interpretability of IBS findings in clinical neuroscience.

## 5. Conclusions

This review highlights how inter-brain synchrony is a condition-sensitive, contextually modulated, albeit nascent marker of interpersonal neural dynamics. It reflects both dispositional and situational components of psychological health and social functioning and has potential to serve as a neural indicator of interpersonal dynamics affected by psychological conditions. Across diverse disorders, IBS provides a window into how individuals connect or fail to connect at the neural level. Its integration into clinical science and practice holds the potential to revolutionize diagnostics, personalize therapy, and ultimately, bridge the gap between brain science and relational healing. Neurochemical mechanisms are plausible modulators of IBS through their direct influence on emotional regulation, cognitive processing, and social behaviors. Future integrative research examining these molecular pathways alongside IBS measures holds promise for clarifying the precise biological underpinnings of altered interpersonal neural synchrony in psychological and neurodevelopmental disorders.

## Figures and Tables

**Figure 1 brainsci-15-01113-f001:**
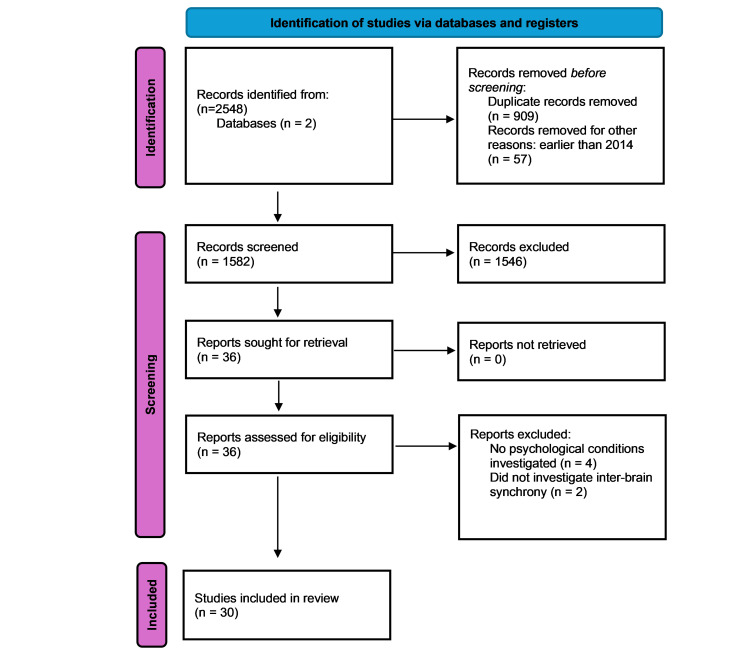
PRISMA flowchart illustrating the selection of studies to be included in this review.

**Figure 2 brainsci-15-01113-f002:**
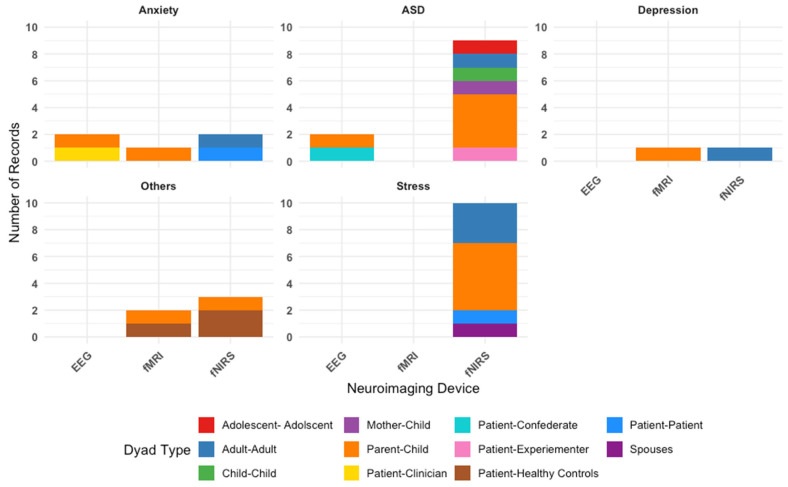
Number of records of inter-brain synchrony studies by type of dyads and type of devices, across psychological conditions.

**Figure 3 brainsci-15-01113-f003:**
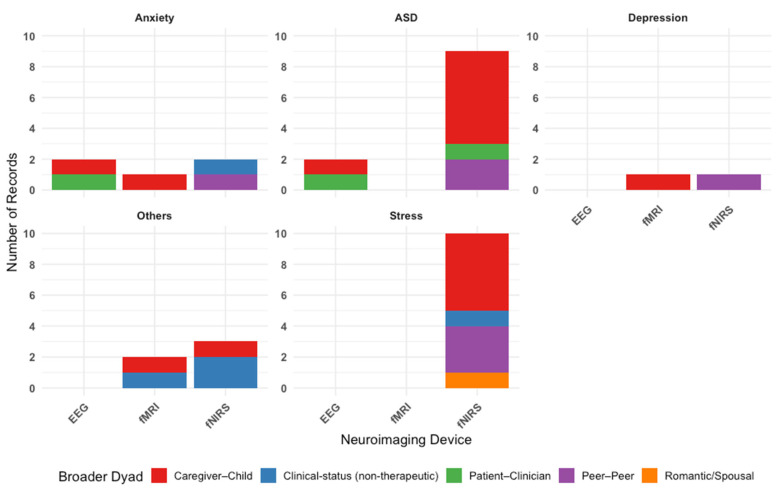
Original dyads (e.g., patient–patient, parent–child, mother–infant, student–student) were consolidated into five higher-order categories to reduce heterogeneity: Caregiver–Child, Patient–Clinician, Clinical-status (non-therapeutic), Peer–Peer, and Romantic/Spousal. Cases with overlapping relationships (e.g., Child–Parent and Child–Therapist) were coded as multi-label. This streamlined taxonomy facilitates clearer synthesis of findings across diverse studies while preserving clinically and developmentally meaningful distinctions.

**Figure 4 brainsci-15-01113-f004:**
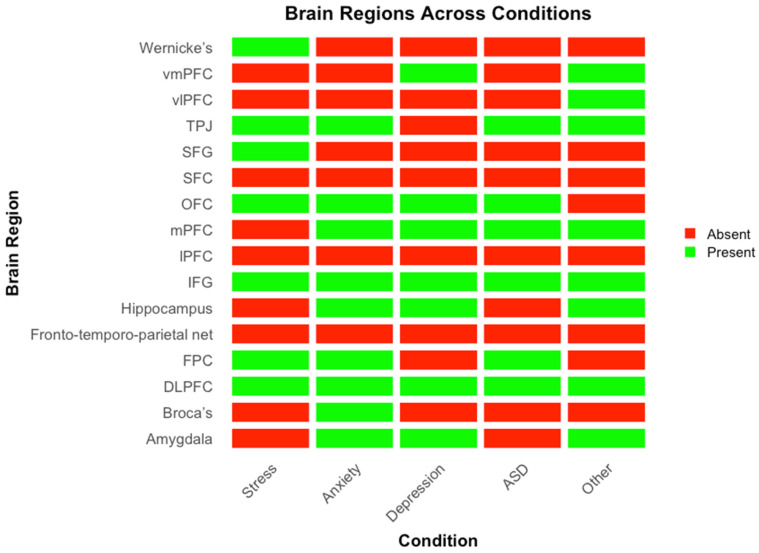
Heatmap showing common brain regions implicated across conditions, with green indicating presence and red indicating absence. Convergent involvement is evident in prefrontal and temporoparietal regions, while limbic structures such as the amygdala and hippocampus appear more selectively in anxiety and depression.

## Data Availability

No new data was created or analyzed in this study. Data sharing is not applicable to this article.

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
