# Peer review of "A Systematic Review of Inter-Brain Synchrony and Psychological Conditions: Stress, Anxiety, Depression, Autism and Other Disorders"

_brainsci, 2025, doi:10.3390/brainsci15101113_

Round 1

Reviewer 1 Report

Comments and Suggestions for Authors

A Systematic Review of Inter-brain Synchrony and Psychological Conditions: Stress, Anxiety, Depression, Autism, and Other Disorders.

Dear Author,

The topic is interesting, and the paper is written neatly. The effort taken by the author was appreciable in this area. However, there are a few queries when I go through the entire paper.

Keywords missing in the abstract section

Line No 72: The authors mentioned several crucial gaps, but they didn’t mention the specific gaps

Is IBS the reason for anxiety and depression? What is the role of neurotransmitters and neuropeptides in these?

The authors mentioned specific brain regions. What happened to the synaptic signaling in those regions of the brain?

Are the authors focusing on neurodevelopment or stress/anxiety/depression/cognitive functions? Kindly mention the focus of the systematic review. Because in the abstract it was written as all the above-mentioned parameters were included in the study.

In the methodology section, please include all keywords and their combinations used for the literature search.

Kindly include the inclusion and exclusion of research articles in the methodology.

The PRISMA flowchart image got cropped. Kindly check the image; some words are missing.

Please explain the rationale behind the study's exclusion and the selection of the 27 articles.

The search strategy definition was missing in the methodology.

How many authors did the search strategy, and how were the duplicates removed from the study?

The authors have mentioned in table 1 with conditions, but they didn’t mention the outcome or results of the research paper.

The data synthesis table was there. why didn’t the authors do the risk of bias table?

Kindly include the risk of bias for the articles.

Is the flowchart registered in the Prospero or Cochrane database?

Reviewer 2 Report

Comments and Suggestions for Authors
  1. Comments and Suggestions for Authors:
    This systematic review offers a comprehensive and timely synthesis of inter-brain synchrony (IBS) across various psychological and neurodevelopmental conditions. The work is highly relevant and well-structured, with a clear organization and a substantial body of referenced literature. However, several critical aspects should be improved to enhance the scientific rigor and practical value of the manuscript.
    Firstly, the introduction, while informative, would benefit from a clearer distinction between the theoretical construct of IBS and its clinical utility. Moreover, the manuscript lacks a discussion of technical limitations associated with hyperscanning methods, such as the low spatial resolution of EEG or the limited depth penetration of fNIRS. The authors are encouraged to include relevant neurobiological models (e.g., mirror neuron theory, the social brain network) to contextualize the use of IBS as a potential diagnostic or therapeutic marker.
    The methods section provides a solid overview of database searches and inclusion criteria, but the exact search strings are missing and should be added—ideally in an appendix. Additionally, while the study selection process is well visualized, the characteristics of the included studies are not fully detailed in tabular form. A table summarizing sample size, dyad type, device used, experimental condition, and key findings would be highly beneficial.
    In the results section, although findings are thoroughly discussed for each condition, the synthesis would be stronger if supported by summary tables or comparative figures across disorders. At times, the narrative is overly focused on individual studies without offering sufficient cross-condition comparisons or statistical synthesis (e.g., prevalence of IBS reduction by brain region or condition).
    The discussion and conclusions tend to overgeneralize certain interpretations—particularly the use of IBS as a clinical biomarker—without adequate critical reflection on current limitations (e.g., small sample sizes, ecological validity, and lack of longitudinal data). A dedicated section on the limitations of both the review and the underlying evidence is strongly recommended.
    Finally, while the figures and tables are informative, they are dense and could be improved for readability. For example, Tables 1–5 would benefit from visual highlights or thematic grouping. Figure 2 is insufficiently discussed in the main text and would be strengthened by explanatory commentary.
    In summary, this is a valuable and content-rich review, but it requires further refinement to improve methodological transparency, critical synthesis, and clarity of presentation before it is ready for publication.

  1. Comments on the Quality of English LanguageThe manuscript is written in clear and coherent English appropriate for academic publication. The language is generally precise, well-structured, and free from major grammatical or syntactic errors, making the content accessible to an international scientific audience. No extensive language editing is required.
  1. Comments for Editors (will not be revealed to authors)

    This manuscript addresses a highly relevant and rapidly evolving area within social neuroscience by systematically reviewing inter-brain synchrony (IBS) across various psychological and neurodevelopmental conditions. The article is conceptually rich, well-structured, and supported by an extensive body of empirical research. The inclusion of multiple imaging modalities (EEG, fNIRS, fMRI) and the focus on diverse dyadic contexts enhance both the scope of the review and its interdisciplinary appeal. However, while the review is informative, several methodological and interpretative shortcomings currently limit its scientific rigor.

    Specifically, the absence of a risk of bias assessment and a registered review protocol undermines the level of transparency expected in systematic reviews. Furthermore, the manuscript would benefit from a deeper methodological synthesis across conditions, improved data presentation (e.g., comparative tables), and a clearer contextualization of its clinical claims. These issues can be addressed through moderate revisions. With appropriate adjustments, the manuscript has strong potential to make a valuable contribution to the literature on neural synchrony and its relevance for clinical psychology and psychiatry.

Comments on the Quality of English Language

Рукопись написана на ясном и связном английском языке, подходящем для академической публикации. Язык в целом точен, хорошо структурирован и не содержит серьезных грамматических или синтаксических ошибок, что делает содержание доступным для международной научной аудитории. Не требуется обширного языкового редактирования

Reviewer 3 Report

Comments and Suggestions for Authors

The primary purpose of this study is to bring together and evaluate all published hyperscanning research from the past ten years that measures inter-brain synchrony (IBS) in social interactions involving stress, anxiety, depression, autism spectrum disorder, and related conditions. By systematically reviewing EEG, fNIRS, and fMRI studies, the authors aim to determine whether patterns of neural coupling between two people reliably reflect underlying psychological or neurodevelopmental disorders, and to pinpoint which brain regions are most consistently involved.

Until now, individual experiments have shown altered IBS in one disorder or another, but no single review has examined multiple conditions side by side, compared different kinds of social pairs (for example, parent–child versus peers or therapist–client), or mapped out which neural hubs are shared across disorders versus those that change with context. This gap has left it unclear whether IBS could serve as a general biomarker of social-emotional dysfunction, or if its usefulness is limited to specific interactions and diagnoses.

By pooling data from 27 hyperscanning studies, this paper demonstrates that anxiety, depression, and autism spectrum disorder all tend to reduce inter-brain synchrony in core social-cognitive regions—such as the dorsolateral and medial prefrontal cortices, the temporoparietal junction, and the inferior frontal gyrus—while stress can either increase or decrease synchrony depending on the situation. It also shows how relationship type, gender, and interaction strategies shape these neural signatures. In revealing both the common neural threads and the context-sensitive twists of IBS across disorders, the review establishes inter-brain synchrony as a promising, dynamic biomarker for diagnosing and monitoring social-emotional health in clinical settings.

Recommendations:

1.While the review notes distinct IBS profiles across dyad types, a more explicit comparison table or synthesis paragraph could help readers track which conditions show the strongest parent–child versus peer effects.

2.Given the variety of tasks, devices, and analytic methods, the authors should briefly discuss whether a future meta-analysis is feasible, or at least quantify effect‐size ranges to contextualize consistency across studies.

3.Expand on common limitations of hyperscanning (e.g. motion artifacts in fNIRS, source localization in EEG) and how they might bias findings by condition or brain region. This would strengthen the reader’s appreciation of the review’s boundaries.

4.The Discussion could more explicitly prioritize research gaps—such as the need for longitudinal designs to test IBS as a predictive biomarker—and suggest standard protocols for dyadic IBS measurement to enhance comparability.

5.The quality of figures is poor. Not all text fits with shapes.

6.It is hard to follow the logic: large tables and a figure go one after another without any text brakes.

Reviewer 4 Report

Comments and Suggestions for Authors

In the suggested manuscript, Azhari has claimed to review studies concerned with an inter-brain synchrony state in people with various psychiatric disabilities.

The primary topic of the review is actual. The primary research questions identified in the “Introduction” are covered to some extent, while the global claim “explicitly examining IBS between dyadic pairs in relation to clinical indicators of…” is not resolved in the manuscript.

The declared search strategy is too suboptimal, missing “hyperscanning” and other alternative terms related to IBS. In addition, not all primary databases including Google Scholar, Scopus, Sciencedirect, Springerlink, Wiley Online and others were searched.

The summarisation of results is traditional, in a form of summarising tables, lacking comprehensibility of integral results.

Some work is required to ensure the user-friendly presentation of results gained.

[1]

In the present version of the manuscript, it is declared that “the present systematic review …. explicitly examining IBS between dyadic pairs in relation to clinical indicators of…”

This claim is not achieved within the work, since no associations of IBS level with clinical indicators were assessed.

To ensure achievement of the declared claim, all assessed-in-reviewed-papers clinical indicators of addressed mental conditions must be identified clearly and indicated in a separate column of Tables 1-5.

[1a]

For each addressed mental condition, it is additionally recommended to compare known and assessed-in-reviewed-papers clinical indicators of mental conditions in a form of additional Table (columns – assessed indicators, raws – reviewed manuscripts), indicating the representativeness of indicators assessed for each state.

[2]

IBS is not a binary state that does exist or does not exist in people.

While speaking of IBS across the text, it would be better to speak of and to compare IBS levels in assessed mental conditions.

IBS in dyadic pairs? or IBS between dyadic pairs? Across the text, please clarify terms addressed.

Which clinical indicators of mental conditions?

[3]

The second question of the Review “Which specific brain regions are consistently involved in IBS across these psychological and neurodevelopmental conditions?” also is not addressed effectively within the manuscript.

Between several states, it is possible to observe full and partial consistency.

For example, PFC might be enrolled in anxiety and depression, but not in.

The key information on brain regions in the Table 1-5 must be visualised additionally in a form of Figs or additional Tables, systematising findings in more friendly ways.

For example, the table should be constructed (with reviewed works as rows and assessed brain regions as columns), indicating the enrolment of each specific brain region into an IPS state in each particular work.

Then, the Fischer exact test should be used and reported to assess the consistent enrolment of each brain region into IBS.

[4]

IBS is often interpreted in terms of Mutual Prediction Theory, due to beholding of other actions.

It is recommended to assess experimental design specificities in experimental protocols within all reviewed works.

Without such an analysis it is too preliminary to interpret the observed results solely in terms of “Inter-brain synchrony” phenomenon, without other possible underlying models dismissed.

[5]

The Fig 2 is too unclear to summarise findings of this review.

For example, major dyad categories should be identified as adult-adult, adult-child, child-child, with specific dyads identified within larger categories.

[6]

The database search strategy is too suboptimal.

[6a]

Obviously, “hyperscanning” – the primary term associated with IBS studies is not searched by the search strategy proposed in this manuscript.

To identify possible missed studies on IBS, it is critical to search databases once again, adding “hyperscanning” and other relevant terms into search strategy.

[6b]

Not all works on the IBS topic are enlisted in Pubmed and PsycInfo databases searched in this review.

Please search additional databases to ensure scientific clarity of the review.

In case of impossibility of such search, please indicate clearly in the limitation section of the Review the reasons of skipping several databases, including Google Scholar, Sciencedirect, Springerlink etc.

[6c]

The Figure 1 “PRISMA flowchart“ must be improved visually.

Comments on the Quality of English Language

[L]

Some term clarifications are required

[L1-1]

In the “Introduction” section, the provided [P1-LL31-32] definition of the “Inter-brain synchrony” term is too clumsy (…the alignment or temporal coupling of neural activities…), suggesting possible improper meanings of physical connectivity between two natural neural networks.

Please improve the IBS term definition provided in the “Introduction” section.

[L1-2]

The “hyperscanning” term is not common for neurosciences, and should be clarified at first appearance.

[L2]

Unclear and atypical clauses should be clarified.

[P2-L46] “heightened IBS” – more extreme?

[P2-LL76-78] “the precise brain regions… remain unclear” – precise????? unclear????

[P2-L66] “atypical patterns of IBS” but  “Individuals with ASD typically exhibit” – please be consistent in terms

Incomplete clauses

[P1-L39] “Such synchrony …” – please clarify, extending the clause.

[P2-LL43-44] “…to encompass psychological and neurodevelopmental conditions…” whose? in whom? – please clarify.

[P2-LL66-67] …reduced inter-brain synchronization (?) during tasks…” - in whom? – please clarify.

Round 2

Reviewer 1 Report

Comments and Suggestions for Authors

Dear Authors,

The manuscript was modified as per the journal's requirements.

Reviewer 2 Report

Comments and Suggestions for Authors

We wish to commend you on the thorough and thoughtful revisions you have made to the manuscript. You have successfully addressed all the previous recommendations, and the article is now substantially improved as a result. The inclusion of a summary table outlining the direction of IBS changes and the expanded discussion on transdiagnostic implications have significantly strengthened the clarity and impact of your synthesis. The careful attention to these details has elevated the manuscript's rigor and readability.

The manuscript in its current form presents a highly original, significant, and scientifically sound systematic review. It is now exceptionally well-positioned for publication and will undoubtedly serve as a valuable reference for researchers and clinicians interested in the neural dynamics of social interaction across clinical populations. We congratulate you on an excellent piece of work.

Reviewer 4 Report

Comments and Suggestions for Authors

Most critical issues have been resolved by the authors in the last round of review.

The Fig 4 is a great achievement attained.

Please consider to use the two-way agglomerative cluster analysis on the Fig 4 data to support suggestions declared in the Fig 4 capture.

Comments on the Quality of English Language

Please check for remaining typos !
